

# Linked within-host and between-host models and data for infectious diseases: a systematic review

Lauren M. Childs[1], Fadoua El Moustaid[2,3], Zachary Gajewski[2,3,4], Sarah Kadelka[1], Ryan Nikin-Beers[1,5], John W. Smith, Jr[4], Melody Walker[1] and Leah R. Johnson[2,3,4,6]

[1] Department of Mathematics, Virginia Polytechnic Institute and State University (Virginia Tech), Blacksburg, VA, USA
[2] Department of Biological Sciences, Virginia Polytechnic Institute and State University (Virginia Tech), Blacksburg, VA, USA
[3] Global Change Center, Virginia Polytechnic Institute and State University (Virginia Tech), Blacksburg, VA, USA
[4] Department of Statistics, Virginia Polytechnic Institute and State University (Virginia Tech), Blacksburg, VA, USA
[5] Department of Mathematics, University of Florida, Gainesville, FL, USA
[6] Computational Modeling and Data Analytics, Virginia Polytechnic Institute and State University (Virginia Tech), Blacksburg, VA, USA

Corresponding author
Leah R. Johnson, lrjohn@vt.edu

## ABSTRACT

The observed dynamics of infectious diseases are driven by processes across multiple scales. Here we focus on two: within-host, that is, how an infection progresses inside a single individual (for instance viral and immune dynamics), and between-host, that is, how the infection is transmitted between multiple individuals of a host population. The dynamics of each of these may be influenced by the other, particularly across evolutionary time. Thus understanding each of these scales, and the links between them, is necessary for a holistic understanding of the spread of infectious diseases. One approach to combining these scales is through mathematical modeling. We conducted a systematic review of the published literature on multi-scale mathematical models of disease transmission (as defined by combining within-host and between-host scales) to determine the extent to which mathematical models are being used to understand across-scale transmission, and the extent to which these models are being confronted with data. Following the PRISMA guidelines for systematic reviews, we identified 24 of 197 qualifying papers across 30 years that include both linked models at the within and between host scales and that used data to parameterize/calibrate models. We find that the approach that incorporates both modeling with data is under-utilized, if increasing. This highlights the need for better communication and collaboration between modelers and empiricists to build well-calibrated models that both improve understanding and may be used for prediction.

## INTRODUCTION

In the study of biological systems, phenomena are often observed at multiple scales—from sub-cellular to entire populations. Here, we focus on the between-host scale and the within-host scale, given their frequent appearance as a dichotomy in the study of infectious disease (*Mideo, Alizon & Day, 2008*; *Handel & Rohani, 2015*; *Willem et al., 2017*). The between-host scale may include how a disease spreads among organisms in a population, while the within-host scale may include intra-cellular or inter-cellular interactions with an invading pathogen. Since biological systems often do not exhibit a clear separation of temporal or spatial scales, there has been increased interest in recent years in how interactions at one scale can affect interactions at the other (*Handel & Rohani, 2015*; *Alizon, Luciani & Regoes, 2011*; *Mideo et al., 2013*; *Theys et al., 2018*; *Dorratoltaj et al., 2017*).

Mathematical and computational modeling, which has a rich history of application to the dynamics of ecological systems and infectious diseases, has been used to study phenomena at both within-host and between-host scales (as well as other scales as is reviewed in *Garira (2017)*). At the between-host scale, classic compartmental models, like the *SIR* model, which represents the interactions between susceptible individuals $S$, infected individuals $I$, and recovered individuals $R$, have been used to predict the spread of infectious diseases between individuals in a population (*Kermack & McKendrick, 1991*, *1927*; *Anderson & May, 1992*). At the within-host scale, models such as the *TIV* model of viral dynamics, which represents the interactions between target cells $T$, infected cells $I$, and virus $V$, have been used to understand viral load within hosts (*Perelson et al., 1996*; *Nowak & May, 2000*).

To understand the outcomes produced by the interactions in and between different scales, a multi-scale model that links the scales may be constructed. For example, an *SIR* model may be used to describe the spread of a viral disease in a population. If the transmission rate between hosts is dependent on the outcome of the viral load from a *TIV* model (since higher viral loads often are associated with higher disease transmission, e.g., *Nguyen et al. (2013)*), the models at the between-host scale and the within-host scale depend on one another, and are thus considered linked. These models can be diverse in their structure and formulation (*Garira, 2017*; *Garira, Mathebula & Netshikweta, 2014*). To be clear, multi-scale models encompass a wide range of possibilities, as reviewed in (*Garira, 2017*). Here, we focus on the within-host and between-host scales for infectious diseases.

Thinking about the implications across scales is important but is also challenging as the relationships are often complex, nonlinear and, therefore, un-intuitive. Previously, theoretical models of multi-scale phenomena have been reviewed (*Mideo, Alizon & Day, 2008*; *Reiner et al., 2013*; *Dorratoltaj et al., 2017*; *Murillo, Murillo & Perelson, 2013*; *Severins, 2012*). Repeated themes of these works and others over the past two decades have included: the need for more data (*Alizon & Van Baalen, 2008*; *Alizon, Luciani & Regoes, 2011*; *Handel & Rohani, 2015*; *Lavine, Poss & Grenfell, 2008*; *Pollitt et al., 2011*); the challenge of integrating scales (*Frost et al., 2015*; *Perelson et al., 1996*;

*Handel & Rohani, 2015*; *Mideo et al., 2013*); and the role of heterogeneity (*Lavine, Poss & Grenfell, 2008*; *VanderWaal & Ezenwa, 2016*). Furthermore, there was an emphasis on the role of particular quantities such as within-host trade-offs (*Martinez-Bakker & Helm, 2015*; *Pollitt et al., 2011*) and immune response factors (*Graham et al., 2007*; *Hawley & Altizer, 2011*).

Of the 22 reviews found by our search, two were themselves systematic reviews (*Dorratoltaj et al., 2017*; *Willem et al., 2017*). The former had a narrow focus on multi-scale models of HIV infections in humans (and returned nine papers), while the latter examined all individual-based models (IBMs) of infectious disease (698 in total) from 2006 to 2015. Of the remainder of reviews, several focused mainly on biological aspects of multi-scale dynamics (*Dolan, Whitfield & Andino, 2018*; *Forrester, Gutiérrez & Coffey, 2016*; *Pimenoff et al., 2018*; *Wilks et al., 2012*), or were more methodologically based (*Kao, 2010*; *Willem et al., 2017*). Two recent works (*Dorratoltaj et al., 2017*; *Theys et al., 2018*) focus specifically on multi-scale models of HIV infections of humans. In particular, *Dorratoltaj et al. (2017)* only returned nine papers, a relatively low number, that is, similar to our own study. However, they actually found three papers that did not appear in our search (*Saenz & Bonhoeffer, 2013*; *Metzger, Lloyd-Smith & Weinberger, 2011*; *Yeghiazarian, Cumberland & Yang, 2013*). As we required within- and between-host components as well as data, none of these three papers match our criteria, and thus it is unremarkable that we did not find these papers through our search.

In this review, we aim to illuminate the state of the field joining experimental data with mathematical and computational models that bridge within-host and between-host scales. By doing so in a systematic manner we expect to identify potential gaps in understanding and methodology. Thus, we examine papers that incorporate models that contain linked within-host and between-host model components as well as explicitly utilize data. While we have related an example that involves the linking of two compartmental models in the context of a viral disease, we do not restrict our search to only compartmental models or to models of viral disease. We find 24 papers which contain both (i) the within-host and between-host scales and their connection and (ii) data. In "Survey Methodology," we describe how we searched for and chose papers. In "Results," we explain trends of the models in the papers we selected. We then conclude in "Discussion" with some overall thoughts on the current literature using multi-scale models with data.

## SURVEY METHODOLOGY

To perform our systematic review we followed the preferred reporting items for systematic reviews and meta-analyses (PRISMA) guidelines (*Moher et al., 2009*). PRISMA is a standard protocol for conducting a systematic review or a meta-analysis. The flowchart showing our procedures are presented in Fig. 1A.

We searched "All Databases" on Web of Science using the search terms *(within-host\* OR in-host-model\* OR among-host-model\*) AND (between-host\* OR nested-model\* OR cross-scale-model\*) AND (pathogen\* OR parasite\*) AND (transmi\*)* for papers published up to December 31, 2018. Terms combined within parentheses with "OR" require at least

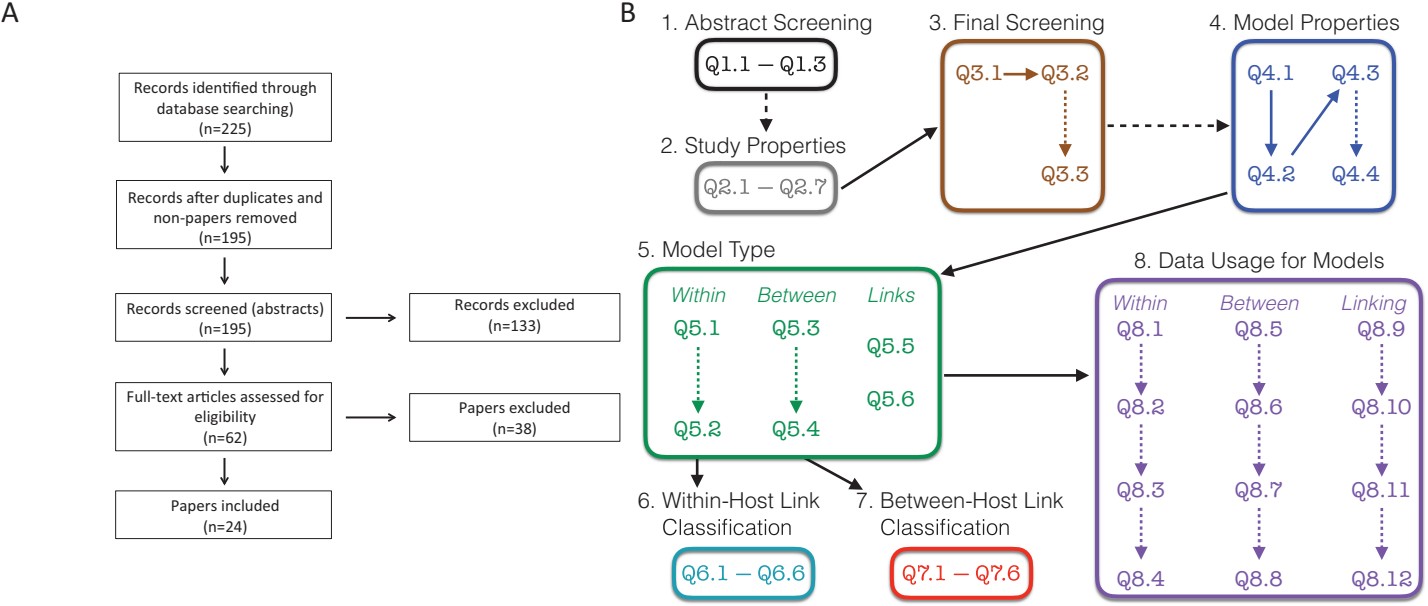

**Figure 1 Schematic of survey methodology.** (A) PRISMA flowchart showing the inclusion of papers. "Non-papers" refers to database entries that were figures or codes. (B) Schematic of the screening and evaluation questions used. Dashed lines indicate links between questions that were conditional, that is, answering the second question/box depended on the answers to the earlier question. For example, details on the study properties (Q2.1–Q2.7) and questions from the final screening stage (Q3.1–Q3.3) were only collected for the 195 papers that were retained following the abstract screening stage. Questions in boxes 4–8 were completed for all 24 papers that remained following the final screening stage. Questions are found in Text S1; Responses are found in Tables S1–S8; References to all included papers are found in Text S2; References to all excluded papers are found in Text S3; All recorded data can be found in our Supplemental Data Sets.

one of these terms while the "AND" between terms in parentheses requires something from each group of terms. Terms that end in a * indicate that any form of the ending of the word would be acceptable. For example, "transmi*" would return papers with transmit or transmission. We note that this search type accesses the "title," "abstract," "author keywords," and "keywords plus." If the search terms are not found in these locations, the papers will not be returned from the search.

Based on these search terms, we obtained 225 results (Fig. 1A). We initially eliminated 29 search results, which included duplicates and other results that were not papers. Further, there was one paper that could not be obtained in English (*Verenini, 1983*); only an Italian version was found. This left us with 195 papers, which we initially screened based on the abstract. Two additional papers were suggested during the review process (*Smith & Mideo, 2017*; *Greenspoon, Banton & Mideo, 2018*), for which the abstracts were screened, and the papers were ultimately excluded. These two papers are included in our presentation of data but do not appear in the survey methodology as they were not recovered by the search.

In the initial abstract screening phase, two randomly assigned people (i.e., two of LMC, FEM, ZG, SK, RNB, MW, or LRJ) separately categorized each of the 195 papers into three categories based on whether it appeared to include a linked model with data based purely on the abstract: "Yes," "Maybe," and "No." A linked model was defined as a mathematical model that includes at least two scales, within-host and between-host, as well

as some explicit link between the scales. The abstract was labeled as follows: "Yes" if it appeared to describe both a linked model with data; "Maybe" if it either (i) clearly described a linked model but was unclear on data, or (ii) clearly referred to data, but was unclear if the included model was linked; "No" if it did not meet any of the above criteria, was obviously a review, or obviously outside the scope of our review. A set of study properties (Fig. 1B, Q2.1–Q2.7) were also collected for each of 158 papers at the abstract screening stage including the focal host species, other mentioned species, the type of infection, and the main results of the paper. Study properties were not recorded for the other 37 papers as they were either review papers not out of scope (19) or deemed out of scope (18). The total number of papers for which we recorded the study properties is 160. This includes the two additional papers that were suggested during the review process. If an abstract was labeled with two "Yes" or with one "Yes" and one "Maybe," we retained the paper for full paper screening; if an abstract was labeled with two "No" we excluded the paper from screening. If an abstract was labeled with one "Yes" and one "No," we reviewed the abstract collectively to relabel it to two "Yes," two "No," or one "Maybe." If an abstract was labeled with one "Maybe" and one "No," the person who labeled "Maybe" was assigned to skim the paper to decide if the paper should be kept or eliminated. If an abstract was labeled with two "Maybe," a third randomly chosen person was assigned to skim the paper to decide whether it should be kept or eliminated. A record was kept if it appeared to have a linked model and/or data, but still was unclear if it had both; the paper was excluded otherwise. Once this process was completed, we kept 62 papers for further screening, and excluded 133 papers based on the abstracts. The reason for exclusion (lacking data, lacking a model, lacking a within-host component, lacking a between-host component, review, or another reason, which needed to be described) was recorded for all 133 papers excluded at this stage (Fig. 1B, Q3.2–Q3.3). Although there may have been multiple reasons to exclude papers, only one reason was recorded.

We then conducted a final screening of the remaining 62 papers by having two individuals (randomly assigned from the full author list) read through the full text of each paper. During this step, a final determination was made for each paper whether to keep it for further analysis or to exclude it. A paper was kept if it contained a linked model with data; a paper was otherwise excluded. The reason for exclusion (lacking data, lacking a model, lacking a within-host component, lacking a between-host component, review, or another reason, which needed to be described) was recorded for all 38 papers excluded at this stage (Fig. 1B, Q3.2–Q3.3). In all, we included 24 papers (*Althouse & Hanley, 2015*; *Chaves, Kaneko & Pascual, 2009*; *Chen, Sanderson & Lanzas, 2013*; *Cooper & Heinemann, 2005*; *Day, Alizon & Mideo, 2011*; *Dennehy et al., 2006*; *Dwyer, Levin & Buttel, 1990*; *Fryer et al., 2012*; *Giardina et al., 2017*; *Hall & Mideo, 2018*; *Handel et al., 2013*, *Handel et al., 2014*; *Kennedy & Dwyer, 2018*; *Leclerc et al., 2014*; *Lindberg et al., 2018*; *McKenzie & Bossert, 2005*; *Mideo et al., 2011*; *Reperant et al., 2012*; *Stephenson et al., 2017*; *Takumi et al., 2010*; *Tuncer et al., 2016*; *Van Dorp, Van Boven & De Boer, 2014*; *Volz, Romero-Severson & Leitner, 2017*; *Vrancken et al., 2014*) in the full analyses (Fig. 1A).
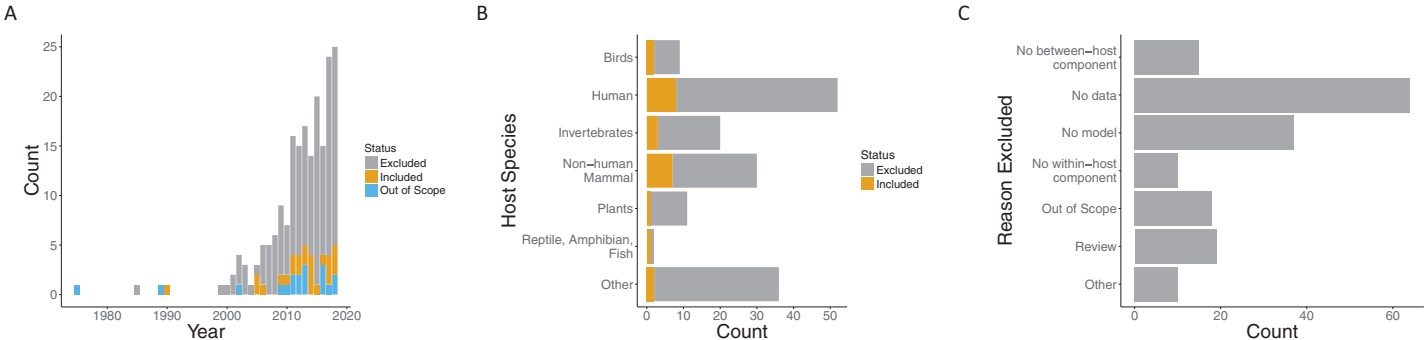

**Figure 2 Summary of papers considered.** Both included and excluded papers by (A) year of publication, (B) host species, and (C) the reason for exclusion (only for excluded papers). Papers were classified as included (gray), out of scope (orange) or excluded (blue) for (A). "Out of scope" designated papers that literally included the search terms but were not topically related.

For the papers that were included, we answered a detailed set of questions, which described important aspects of the model (such as the transmission route), how the models were linked, how the data was used in the model, etc. (Text S1; Fig. 1B, Q4–Q8). We further characterized the journal in which each paper appeared as a general audience journal, a specialized biological journal, a primarily mathematical/computational journal, or a biology sub-discipline journal (Fig. S1).

## RESULTS

### Traits of included compared to excluded papers

Our search process (including suggestions in the review process) yielded 197 papers published over the span of more than 30 years. While the earliest included paper was from 1990 (*Dwyer, Levin & Buttel, 1990*), the next papers that met our requirements were published 15 years later (*Cooper & Heinemann, 2005*; *McKenzie & Bossert, 2005*). In the interim, a few more papers were published, but interest in this general area grew quickly starting in 2005. Both the number of papers loosely related to the topic (i.e., those excluded) and papers meeting our criteria to include both models and data (i.e., those included) increased in that time frame (Fig. 2A).

Papers spanned a variety of host species systems (Fig. S2). Infections of humans were, not surprisingly, the most common in both the excluded (44/135) and included categories (8/24), followed by non-human mammals (30 overall) and invertebrates (20 overall). Although human infections were considered in the largest number of included papers overall, the proportion of included papers when broken down by focal host species is largest for non-human mammals (7/30) and approximately the same for invertebrates (3/20) and humans (8/52). The most common reason for exclusion was a lack of data being used with the model (64/135) followed by no model (37/135) (note, that only one reason was recorded for each paper). That is, many papers explore within- to between-host transmission either from a modeling or empirical perspective, but many fewer link the models robustly to data. Recently, there have been a number of review papers on multi-scale models with data, another common reason for exclusion (22/135).
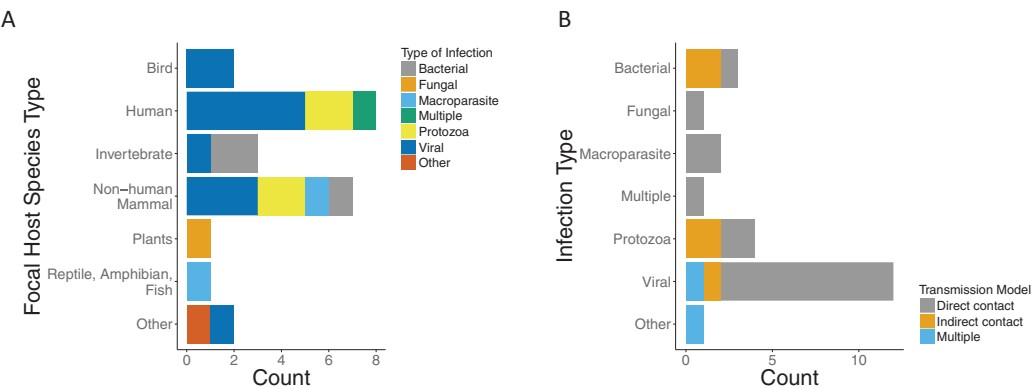

**Figure 3 Focal host species and infection types for included papers.** (A) Type of infection across host species for included papers as bacterial (gray), fungal (orange), macroparasite (blue), multiple (green), protozoa (yellow), viral (dark blue), or other (red). (B) Modeled transmission route across infection types for included papers as direct contact (gray), indirect contact (orange) or multiple routes (blue).

## Traits of included papers

We considered whether the aim of each paper was primarily strategic (trying to understand underlying dynamics) or primarily tactical (trying to make predictions) (*Nisbet & Gurney, 1982*). Of the papers examined, most were classified as primarily strategic and very few papers as primarily tactical. Only one paper was classified as both strategic and tactical (*Vrancken et al., 2014*) (Table S4.2). Included papers were rarely found in highly specialized non-mathematical journals (2/24), but were relatively equally spread between mathematically focused journals, biology focused, and for a general audience (Fig. S1).

### *Infection, host, and transmission categorization*

We found that the majority of the papers and models focused on a single infection. Most infection types were viral (12/24), with protozoa being the second most common (4/24). The host species were predominately mammals (15/24), of which eight were human hosts (Fig. 3A). Most papers modeled transmission as direct contact across infection types. Half of those with protozoa infection type (2/4) and two thirds of those with bacterial infection (2/3) were modeled by indirect contact. Of the papers which modeled viral infections, all but two model direct transmission. In addition, there was one which was indirect (*Handel et al., 2014*) and one with multiple modes of transmission (*Handel et al., 2013*) (Fig. 3B).

### *Model characteristics*

The multi-scale models reviewed are composed of three parts: the within-host model, the between-host model, and the linking mechanism. Although we only considered papers with models including all of these three components, as well as data, the papers varied on the focus of their results. Approximately two-thirds (15/24) of papers primarily investigated how the within-host dynamics affect the between-host dynamics; only two papers focused on the impact of the between-host dynamics on the within-host dynamics. The remaining third

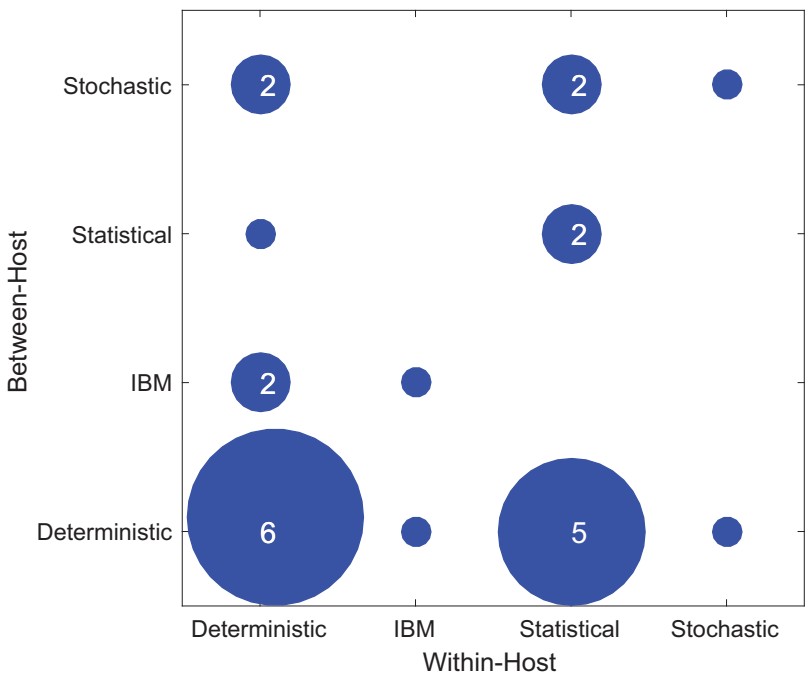

**Figure 4 Types of modeling framework used in included papers.** The *x*-axis shows the model types used in the within-host part of the model while the *y*-axis shows the model types used in the between-host model. The dots' diameter represents how many papers used a particular framework.

of the papers either examined both of the above directions of impact (i.e., how the within-host and between-host dynamics effect each other) or the influence of within- or between-host dynamics on another factor in their model (Table S4.3).

In the multi-scale models we considered, the within-host component and between-host component were both modeled explicitly. We characterized each of the within-host and between-host models used as either a deterministic model, a stochastic model, an IBM, or a statistical model. In all cases, our categorization is based on the process/mechanism portion of the model being considered. Deterministic models included any mechanistic/process based model that did not include stochasticity in the process. For example, this may include ordinary differential equation (ODE)/compartmental models. These models may have been fit assuming a stochastic observation model overlaid on the dynamics. Stochastic models include any model that includes stochasticity in the description of the mechanism/process. Examples of this would include stochastic SIR models, or stochastic differential equations, but *not* IBMs. Here, IBMs include any models in which individuals or agents were modeled separately from each other and allowed to interact within a simulated environment. This is in contrast to models that include data collected on individuals and where these are used to parameterize models (we would refer to these as "trait based"). IBMs as defined here may contain both deterministic and stochastic components. These models are typically much more difficult to analyze and fit to data than either of the other two flavors of mechanistic models, hence why we considered them separately. Finally, statistical models are those that seek to fit a function to data without the

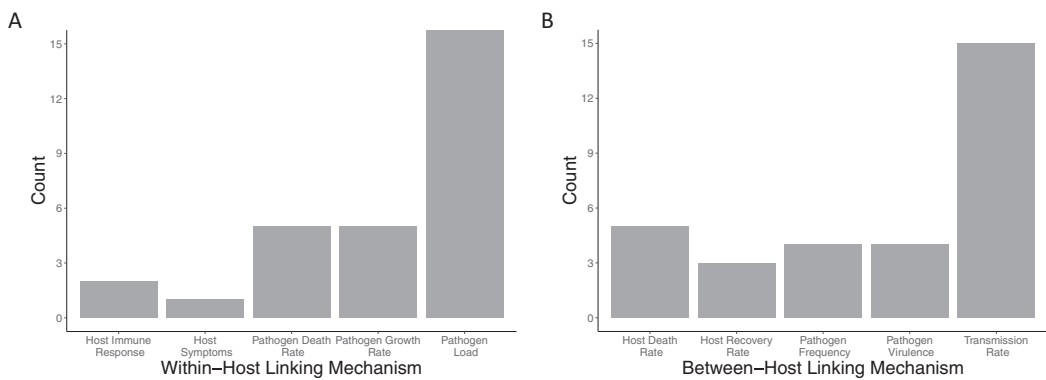

**Figure 5  Mechanisms used to link between and within-host models together.** The number of included papers that used the each of the (A) within-host linking mechanisms and (B) between-host linking mechanisms to connect the models together.

function having an explicit link to a mechanism for what produces a pattern. This would include all flavors of regression, and many machine-learning approaches.

Figure 4 shows the types of within-host and between-host models used in the included papers. Most studies used the deterministic model type at least once, either for within- or between-host models and sometimes for both. In the included papers, within-host models were most commonly deterministic (11/24), followed by statistical (9/24), individual-based (2/24), and stochastic (2/24). In contrast, for the between-host models, the majority were deterministic (13/24), with a lower and more evenly distributed representation of stochastic (5/24), individual based (3/24), and statistical (3/24). One study used an IBM model type for both the within-host model and the between-host model (*Van Dorp, Van Boven & De Boer, 2014*). In general, studies did not typically use the same modeling approach for both the within- and between-host components. As for host type, there was no evident correlation between model types and the focal host species used in the model (Fig. S3).

Within- and between-host models can be linked in three different ways: within- to between-, between- to within-, or bidirectionally. Regardless of the varied emphasis of the results of the papers, as described above, we examined the formulation of the linking mechanism of the model. Among the included papers, 12 of the studies linked the within-host model to the between-host model while 12 linked bidirectionally—both within-host to between-host and between-host to within-host (Table S5.5).

To link the within-host and between-host models, a linking mechanism was needed, which we categorized either as a state or a trait. Linking via a state meant that an outcome of the model was used; for example, the pathogen load at the within-host scale or the number of infected individuals at the between-host scale. In contrast, a trait was a parameter of the model; for example, the pathogen growth rate at the within-host scale or the transmission rate at the between-host scale. The model framework was categorized in one of three ways: linked only by states, only by traits, or by both traits and states. Furthermore, models could also have multiple linking mechanisms. In the included papers, eleven studies used state variables, four used trait variables, and nine used both (Table S5.6).

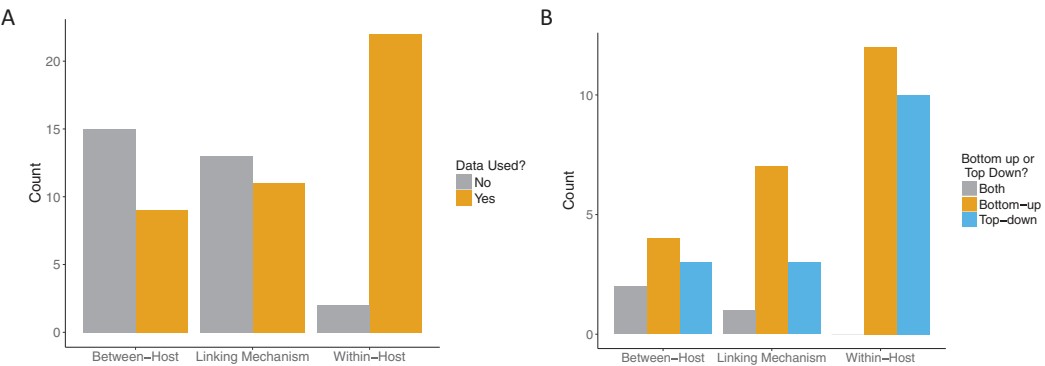

**Figure 6  Role of data in multi-scale modeling efforts.** (A) Scale (within-host, linking, or between-host) at which data was incorporated (orange) in the multi-scale models. Some models used data at more than one level. (B) How the data was incorporated into the models: bottom-up, that is, fitting traits (orange); top-down, that is, fitting states (blue) or both (gray).

Within-host models (Fig. 5A) are linked to the between-host models mostly via the pathogen load, with more than half the papers using this linking mechanism (18/24). Pathogen growth rate was the second most used trait to link the within- to between-host model (5/24 papers). All other within-host linking mechanisms were used in two or fewer papers. Between-host models were also linked into the within-host models (Fig. 5B) based on primarily a single trait, the transmission rate (15/24). All other between-host linking mechanisms were used in five papers or fewer.

### Role and method of data incorporation

All papers that passed the screening criteria utilized data in at least one component: the within-host component, the linking mechanism, or the between-host component. Our threshold for data use for model fitting was quite high. We typically required that data be explicitly fitted to some component of the model, for example, fitting the "state" of the system (e.g., the number of observed infectious individuals in an SIR model) to case data. However, one included paper did not explicitly fit data. Instead *Kennedy & Dwyer (2018)* constructed sophisticated linked within and between host models for baculovirus in gypsy moths and then compared the predictions of the calibrated model to newly obtained experimental data in a validation step. Even among the relatively small sample of papers that included data at all in these multi-scale models, most did not use it for more than one scale of their model (Fig. 6A). While most of the included papers (22/24) used data at the within-host scale, only seven papers used data at both the within-host and between-host scales, of which only four also used data for the linking mechanism. Papers that included data for both the linking mechanism and the between-host scale also included data for the within-host scale.

Across all model scales, bottom-up, that is, fitting of traits, was utilized more than top-down, that is, fitting of states, or other methods (Fig. 6B). Only one paper used a mixture of bottom-up and top-down data fitting methods (maximum likelihood and least squares, respectively) at different scales, although one paper did not specify explicitly how the data was incorporated into the model (*Cooper & Heinemann, 2005*). For data fitting that

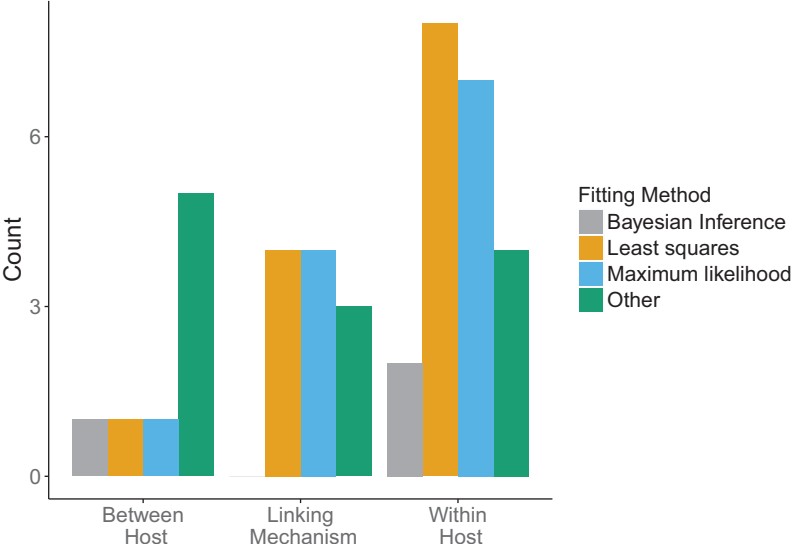

**Figure 7 Method used in data fitting at each scale.** Three fitting methods were considered: Bayesian inference (gray), least squares (orange), maximum likelihood (blue). All other fitting methods were included under "Other" (green). Different fitting methods could be used in the same papers for different scales.

was bottom-up, the majority of papers (5/12 within-host, 3/7 linking mechanism) used least-squares or maximum likelihood (6/12 within-host, 4/7 linking mechanism, 1/4 between-host). Bayesian inference, although a popular statistical method, was only used three times in the included papers (Fig. 7). Only a single paper (*Volz, Romero-Severson & Leitner, 2017*) recorded using multiple fitting methods at the same scale, and most papers used the same fitting method across all scales.

There was a diversity of fitting methods used across scales (Fig. 7). Different fitting methods could, and often were, used in the same papers for incorporation of data at different scales. The least squares method was used most when fitting data at the within-host scale, followed by maximum likelihood. Similarly, linking mechanisms primarily used least squares and maximum likelihood to fit data. In contrast, between-host models were less consistent. Across model components the category "Other" was mostly comprised of qualitative fitting methods, or papers for which authors were imprecise about how they fit the data, but it also included methods such as cubic-spline interpolation.

# DISCUSSION

Our objective in this review was to determine how multi-scale infectious disease models, focusing on within-host and between-host scales, are used when they directly incorporate data. We focused on which host species are modeled, which pathogens are modeled, which types of models are used, how the within-host and between-host dynamics are linked, and at what scale data has been used. We found that it was most common for these models to describe a human population, to model a viral disease, to use a deterministic model at either of the two scales considered, to link the pathogen load at the within-host scale, to link the transmission rate at the between-host scale, and to use data at the within-host scale. It was

least common for these models to describe a plant, fish, reptile, or amphibian population, to model a bacterial, macroparasite, or fungal infection, to use a stochastic model at either of the two scales considered, to link host symptoms at the within-host scale, to link the host recovery rate at the between-host scale, and to use data at the between-host scale.

We speculate on the reasons for these outcomes. As human disease has tangible consequences directly impacting the wider population, it is unsurprising that the primary host species to examine these multi-scale interactions was found to be humans. However, the importance of other species both economically and ecologically leaves the door open for further study of these interactions. The dominance of viral disease as the focal pathogen likely results from the rich history of mathematical modeling in viral disease as well as their prominence in the human community. In choosing which type of model to use, deterministic models do not include the mathematical and computational complication of stochasticity, making them often easier to simulate and analyze than stochastic or IBMs. Further, explicitly linking the within-host and between-host scales is challenging. Many studies defaulted to the standard assumption that a higher pathogen load often correlates with a higher chance of disease transmission, making pathogen load the simplest way to link the within-host and between-host scales. Other linking mechanisms are often difficult to model because there may not be an obvious relationship in how two elements at different scales affect one another. The incorporation of data was primarily at the within-host scale, perhaps stemming from the fact that some of these relationships can be obtained through laboratory-based research. In contrast, between-host data may often require large-scale resources and monitoring.

We were quite surprised that our search yielded only 24 papers that included both across-scale modeling and substantial use of data. It is possible that our particular search terms may have been overly restrictive. For instance, the search term "pathogen" may be less likely to be used to describe infectious macro-parasites (e.g., worms). Nonetheless, our relatively small included set indicates that there is considerable scope for further work to be done in the area of data-rich multi-scale modeling of infectious diseases. Given the specific results of our review, we propose that future research could productively focus on (i) exploring alternative linking mechanisms and (ii) incorporating more and varied data at all scales.

Most studies we reviewed appeared to use the simplest assumption to link the within-host and between-host scales, namely, linking the pathogen load at the within-host scale to the transmission rate at the between-host scale. While this assumption may be appropriate for some diseases, there are other potential mechanisms that could be used to provide links between scales, separately or perhaps in addition to pathogen load. For example, these could include how host immunity affects the transmission rate or how pathogen load affects the pathogen virulence among the population. The possibility of using immune factors to link multi-scale models has been raised as a potential simplifying method (*Graham et al., 2007*; *Hawley & Altizer, 2011*). Within-host data on antibodies, when available, could be used as a measure of host immunity. Such simplifications, however, raise the question of which immune components are sufficient. Another major barrier, particularly in wild populations, is the lack of validated assays (*Hawley &*

*Altizer, 2011*). Accounting for these interactions could produce models that make complementary or potentially divergent predictions of transmission outcomes, and in turn be used to elucidate the effects different treatments have on disease spread.

The lack of data was the major reason that our search only uncovered 24 papers (Fig. 2C). Thus, a major gap in bridging within-host infection dynamics and between-host transmission is the existence and incorporation of data. This appears not to have improved significantly since similar observations in 2008 (*Alizon & Van Baalen, 2008*) and in 2015 (*Handel & Rohani, 2015*). Although there are cases where appropriate data for a model does not currently exist and must be collected in a new experiment, a greater effort should be put forth to work with and incorporate existing data sets. This is especially true at the population level, where data are particularly difficult and expensive to collect. Far more of the papers we examined included data at the within-host scale (Fig. 6A), likely due to the accessibility and scale of data that can be collected in a lab setting. Along with more data overall, the incorporation of more varied data at a variety of scales will enhance the utility of multi-scale disease modeling.

## CONCLUSION

Important results about disease spread can be gleaned from modeling the interactions at both the within-host and between-host scales. While current research has mainly focused on simple assumptions, we believe that including additional complexities in future models may help to better explain observations from the field. Multi-scale modeling provides a great opportunity for empiricists and theorists to work together, and to contribute to the understanding of the drivers, treatments, and control of infectious disease.

### Funding
The authors received no funding for this work.

### Competing Interests
The authors declare that they have no competing interests.

### Author Contributions
- Lauren M. Childs conceived and designed the experiments, performed the experiments, analyzed the data, authored or reviewed drafts of the paper, approved the final draft.
- Fadoua El Moustaid conceived and designed the experiments, performed the experiments, analyzed the data, prepared figures and/or tables, authored or reviewed drafts of the paper, approved the final draft.
- Zachary Gajewski conceived and designed the experiments, performed the experiments, analyzed the data, prepared figures and/or tables, authored or reviewed drafts of the paper, approved the final draft.
- Sarah Kadelka conceived and designed the experiments, performed the experiments, analyzed the data, authored or reviewed drafts of the paper, approved the final draft.

- Ryan Nikin-Beers conceived and designed the experiments, performed the experiments, analyzed the data, authored or reviewed drafts of the paper, approved the final draft.
- John W. Smith, Jr performed the experiments, analyzed the data, prepared figures and/or tables, authored or reviewed drafts of the paper, approved the final draft.
- Melody Walker conceived and designed the experiments, performed the experiments, analyzed the data, authored or reviewed drafts of the paper, approved the final draft.
- Leah R. Johnson conceived and designed the experiments, performed the experiments, analyzed the data, authored or reviewed drafts of the paper, approved the final draft.

## Data Availability

The raw data (files with the assessments of each paper) are available as Supplemental Files.

## Supplemental Information

Supplemental information for this article can be found online at http://dx.doi.org/10.7717/peerj.7057#supplemental-information.

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
