# Peer review of "Linked within-host and between-host models and data for infectious diseases: a systematic review"

_PeerJ, doi:10.7717/peerj.7057_

## Round 0.1 · original submission · Major Revisions

While I have given a decision of "Major Revisions" I actually think the work required to alleviate the reviewer's concerns will be relatively easy (I just wanted to give you the extra turn around time, if needed).

You'll notice that two reviewers (#2 and #3) suggest expanding the search terms you used, in hopes of generating a larger and more representative sample. I am in favor of that suggestion. Reviewer 2 mentions searching sources other than Web of Science, but this concerns me less; feel free to expand the search to other sources, but my take is that expanding the search terms is more important, and perhaps easier to do at this stage.

Reviewer #3 also mentions some specific studies that might have been missed by your search for various reasons. Based on your own knowledge of the field, are there other papers that you know were missed for various reasons? If there are few, maybe they are worth mentioning? I don't suggest you include them in your data set, but they might be worth mentioning in the Discussion. I leave this up to you, however.

The reviewers all make good points aimed at clarification. I urge you to seriously consider their comments. Note that reviewer #1 has submitted a PDF with comments.

·

Basic reporting

The manuscript is well written. However, there is no sufficient field background/context provided. The article does not include sufficient introduction to demonstrate how multiscale models which integrate within-host and between-host scale fit into the broader field of multiscale modelling of infectious disease systems. Infectious disease systems have many levels of organization apart from the host level. The authors should have a more general introduction that discusses multiscale modelling at the various levels of organization (cell level, tissue level, host level, etc.) and then indicate why they chose multiscale modelling at host level.

Experimental design

This is a review paper. The research objectives are well defined, relevant & meaningful. The paper is within the aims and scope of the journal. The study design is described in sufficient detail.

Validity of the findings

The findings of this study are clear. the conclusion are well stated, linked to original research question. However, the authors should justify the conclusion that host immunity can be considered as an alternative linking mechanism in multiscale modelling of infectious disease systems.

Additional comments

More general comments are in the pdf file attached.

Reviewer 2 ·

Basic reporting

see below

Experimental design

see below

Validity of the findings

see below

Additional comments

This is a nice review worth publishing, but I have several concerns that I think should be addressed.
My main concern is about the search strategy. I find it too narrow to be a convincing systematic review. First, it seems to me your search terms are not broad enough. For instance I have seen papers that use the term “in-host” instead of “within-host”. Similarly, there might be papers who have a population-level component but might not use the word “between-host”. With your search terms, you seem to be missing those. Further, systematic reviews generally search across multiple sources, it is unusual for systematic reviews to only search a single source (despite WoS being more than one database, I consider it essentially a single source). I recommend you re-do your search across multiple sources (e.g. include Scopus or others that you think might be complementary to WoS) and use somewhat more flexible search terms to ensure you get all relevant studies, not only those who explicitly use the term “within-host”. Such a broader search would make me more confident that this was truly systematic/broad.

My remaining comments are less substantial, listed in order of appearance:

Abstract: calling within-host the 1st scale is a bit arbitrary. One could equally say that within-cell/intracellular is the 1st scale. So I would re-word some to indicate that here you are looking at 2 of the – possibly many – ways one can define scales. Start of introduction is worded better.

Line 25: typo, should be holistic

Line 87: I’m unclear what ‘out of scope’ refers to. Can you clarify? It seems to me that if it’s not a multi-scale model and/or doesn’t contain data, it is flagged as ‘No’ for those categories. What other ‘out of scope’ evaluation did you perform? Where there some papers that were multi-scale models with data but you ignored them? If so why? Needs some more explanation.

Figure 3: since there are only 2 “other” in 3A and 1 in 3B, maybe explicitly state what this ‘other’ is. Especially for infection type, I’m wondering what it could be that’s not one of the listed categories.

Figure 4: I find that figure confusing since it looks as if these are mutually exclusive categories when they are not. IBM could be either deterministic (rare) or stochastic (common), any deterministic or stochastic model could be fit to data, which would make it ‘statistical’. The only clear mutually exclusive categories are deterministic vs stochastic. I think you need better terminology for this. I assume what you mean by deterministic is also a mechanistic, compartmental model (i.e. ODE), and stochastic is also a mechanistic compartmental model, while IBM is individual based (and maybe deterministic, maybe stochastic), and what you call statistical I assume by that you mean a non-mechanistic/process/system model?

Line 153 and following and 172 and following. I’m not clear how these paragraphs describe different things. And in the 1st paragraph, you say that one paper links between to within, while on line 175 you say that no paper linked between to within. That’s confusing, I can’t follow. Can you please clarify?

Line 191 and following. Could you be a bit more explicit about data use? The statement that one paper didn’t explain how they used data puzzles me. I was under the impression up to that point that all papers using data ‘used’ them in some statistical/fitting/inference way. Or do you also include papers that only parameterize their models based on data, without any statistical fitting? I would like a bit more explanation on this. Also, most of your analyses (i.e. up to this paragraph) could have been done on all papers, not just the ones with data. To justify excluding all those and focusing on the ones with data only, it seems to me you need to put some more focus on the data part, i.e. add further analysis/discussion of the connection between models and data. Maybe a version of figure 4S, together with more exploration on the model-data link, might be suitable for the main part of the paper?

Supplement:
• You already include the flowchart in the main text, why include again?
• Your pdf document of excluded papers also includes all the included papers.

Reviewer 3 ·

Basic reporting

In general, I really appreciate the theoretical and applied motivation for this study. I also feel that the study is timely and holds potential for applied applications of the questions raised in the paper. The paper is overall well written, thoughtfully constructed, and addresses important questions. I think the paper would benefit enormously from only a few very minor modifications.

The paper is well written, thoughtfully constructed, addresses an important question, and yields interesting and timely results. I provide a few comments and suggestions, I hope that they will prove constructive.

Experimental design

Lines 76-77: I appreciate the rigorous approach taken in the systematic review. I can’t help but wonder if your review would be expanded if the following search terms were included: cross-scale models, nested models (e.g., Gilcrest was an early proponent of these models and he uses the term nested models), in-host, among-host. In addition, I am assuming that the studies covered in the previous reviews on this topic that you mentioned were also included in your study (i.e., even if they did not appear in your Web of Science search)? For example, Hite and Cressler 2018 and Greenspoon et al., 2018, Smith and Mideo 2016 are a few examples that I do not see included in your reference list (i.e., in any of the main text or online supplementary information). I understand that this may seem like harping on a few studies not being included and I do not mean to do that. It’s just that a large portion of the current manuscript is devoted to describing the search terms. Given this emphasis, I just wonder if more studies would be included in the search terms were expanded.

Validity of the findings

In this manuscript, the authors conduct a systematic review of the literature to examine the current state of cross-scale models that integrate data into the theoretical analyses. They provide a detailed overview of the specific types of models, host-parasite systems, linking mechanisms, and sub-fields addressing these questions. Their review illustrates that despite repeated calls for data-theory integration, few studies currently meet these demands.

This study reports valid findings that appear quite timely and useful across multiple fields.

Additional comments

Lines 54-55: Are there references supporting the relationship between vial load and transmission? I know these relationships are quite difficult to obtain empirically and while this relationship is often assumed, I was not aware that there are many empirical studies that explicitly quantify this relationship. If so, providing those references would be quite useful. If not, maybe revise the sentence?

---

## Round 0.2 · accepted · Accept

Lauren and Leah,

Thank you for you patience during my final, positive, evaluation of your manuscript. I have just a few things for you to consider before uploading your final version:

1. Consider adding in the main text that you also search Scopus and found no extra papers (if a reviewer was curious, perhaps other readers will be, too.

2. Consider adding citation information for all 24 included papers in your reference section. I like to see papers included in a review get "rewarded" with a citation, and since there are only 24 and PeerJ is online, this shouldn't impact your page count and publishing costs (I think). However, PeerJ may have restrictions on including citations in the bibliography that are not cited in text. I've added a note about this for PeerJ staff along with my acceptance notice.

Bear in mind that the above are merely suggestions. The final decision is up to you.

Best,
Andrew